

# A statistical method to validate reconstructions of late-glacial relative sea level – Application to shallow water shells rated as low-grade sea-level indicators

Milena Latinović[1,2], Volker Klemann[1], Christopher Irrgang[1], Meike Bagge[1], Sebastian Specht[1,3], and Maik Thomas[1,2]

[1]GFZ German Research Centre for Geosciences, Potsdam, Germany
[2]Free University, Berlin, Germany
[3]University of Potsdam, Potsdam, Germany

**Correspondence:** Milena Latinović (milena.latinovic@gfz-potsdam.de)

**Abstract.** In this study, we propose a statistical method to validate sea-level reconstructions using geological records known as sea-level indicators (SLIs). SLIs are often the only available data to retrace late-glacial relative sea level (RSL). Determining the RSL from SLI height is not straight forward, the elevation at which an SLI was found usually does not represent the past RSL. In contrast, it has to be related to past RSL by investigating sample's type, habitat and deposition conditions. For instance, water distribution at which a specific specimen is found today can be related to the indicator's depositional height range. Furthermore, the precision of dating varies between geological samples, and, in case of radiocarbon dating, the age has to be calibrated using a non-linear calibration curve. To avoid an a-priori assumption like normal-distributed uncertainties, we define likelihood functions which take into account the indicative meaning's available error information and calibration statistics represented by joint probabilities. For this conceptional study, we restrict ourselves to one type of indicators, shallow-water shells, which are usually considered as low-grade samples giving only a lower limit of former sea level, as the depth range in which they live spreads over several tens of meters, and does not follow a normal distribution. The presented method is aimed to serve as a strategy for glacial isostatic adjustment reconstructions, in this case for the German Paleo-Climate Modelling Initiative PalMod (https://www.palmod.de/en) and by extending it to other SLI types.

## 1 Introduction

The reconstruction of past relative sea level has been the scope of interest of numerous studies over the years (Kopp et al., 2009; Dutton and Lambeck, 2012; Vaughan et al., 2013). Thus, a general approach has been set, where, in situ sediments, fossil organisms, morphological and archaeological features governed by the paleo-sea level, are considered to be reliable estimates of past changes (Shennan et al., 2015). These elements are categorized as sea-level indicators (SLIs). Each SLI contains four main attributes; location of the sample on the Earth, deposition age, elevation usually related to present sea level, or an ordinance datum and the tendency, if the sample allows specification if sea level was rising or falling during the time of deposition. Some studies refer to SLIs as observational data of former sea level, but this term is not suitable since SLIs



are geological samples. In addition to leveling and dating models, the actual sea level has to be derived from the indicative meaning based on the living environment or the deposition conditions of the sample/specimen. Therefore, in the Handbook of Sea-Level Research (Shennan et al., 2015), the term proxy reconstruction is suggested for SLIs instead.

The majority of SLIs representing the last 20,000 years in formerly glaciated regions are dated using the radiocarbon method with high confidence limits (95%) in modern studies. It is important to mention that a change in relative sea level (RSL), which measures the sea level with respect to the Earth's surface, is defined as a change relative to present day. According to the convention for Holocene and Quaternary timescales, age is measured in years before present (yr BP) which is equal to AD 1950.

Unfortunately, only few sea-level indicators form at the exact height of paleo RSL, or, are deposited directly above or below sea level. The majority of SLIs was deposited at some water depth or above the shore line. Therefore, the measurement of sample's height related to modern RSL is giving us only the "indicative meaning". This term first appeared in van de Plassche's Manual for Sea-level Research (1986), which was a part of IGCP Project 61 and 200, one of the first projects dealing with sea-level change since last de-glaciation. Van de Plassche (1986) explained that SLIs feature indicative meaning with respect to present day RSL, below or above it, and that direct measurement of sea level gives instantaneous or mean sea level, while

indicators provide derived sea level.

    Van de Plassche was a pioneer in standardizing the data due to growing interest of studies for SLI databases. Later on, in the Handbook of Sea-Level Research, a database format is suggested in order to more quantitatively analyze the quality of indicators (Hijma et al., 2015). Effort was made from several studies to compile open-access databases due to the growing need for reliable data for models reconstructing palaeoclimate, and to exploit more rigorous inventions of indicator compilations

(PALSEA working group). Special need for SLIs is identified in glacial isostatic adjustment models (GIA), where SLIs serve as model constraints for the inference of mantle viscosity and glaciation history (e.g., Peltier and Andrews, 1983; Nakada and Lambeck, 1991; Mitrovica and Peltier, 1992; Milne et al., 2005; Peltier et al., 2015). Data interpretation is subjective, but it should always include measurement errors and limitations (Düsterhus et al., 2016). Different methods for data interpretation can be found in the literature. In the studies shells are usually only considered as representation of the lower limit of former

relative sea level. Wolf et al. (2006) and Klemann et al. (2007) used fuzzy logic to formulate a classification scheme of the deposition conditions, which served for systematic interpretation of limiting points as well as index points. Kopp et al. (2009) suggested a censored normal distribution for limiting points in order to derive the posterior probability distribution of sea level and ice volume through the last interglacial based on SLIs. Hibbert et al. (2016) suggested a method to exploit the depth distribution of coral's living environments to overcome the problem of specimens, which are situated in a depth range with

respect to sea level. And recently, Caron et al. (2018) suggested to associate a Heaviside distribution to such limiting data. In this study, we partially follow the work of Hibbert et al. (2016) and investigate whether we can improve the indicative meaning of shallow-water shells by using a more rigorous statistical method, where the depth range of living shells is respected. This method is applied in the Hudson Bay region since it can be compared to a number of previous studies focusing on the same region (Wolf et al., 2006).



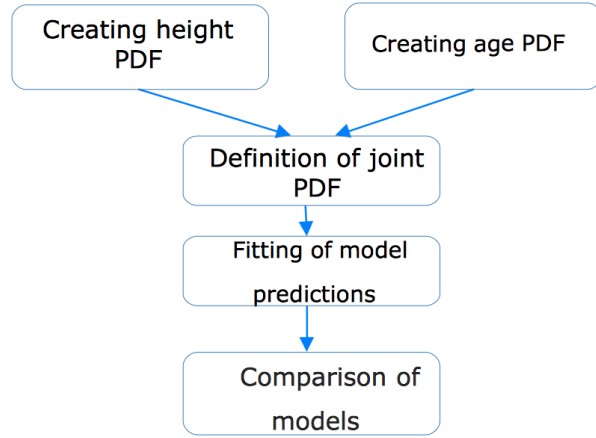

**Figure 1.** Flowchart of model workflow.

The study is set up as follows: first we derive joint probability from height and dating information (Sec. 2.1, 2.2, 2.3), then we determine the conditional probability of model predictions (Sec. 2.4), and apply the method to constrain a given model ensemble of a GIA reconstruction for the Hudson Bay as a prominent example (Sec. 3).

## 2    Methodology and data

The goal of the statistical model we propose is to analyze indicative meaning of different SLI types and to apply the method
for constraining model ensembles in GIA and Earth system modeling. In the following sections, we describe the different steps to our data modeling strategy (Fig. 1).

The statistical model we propose is based on Bayes' statistics, where joint probability density functions (PDF) are derived from the height information and age information of a specific SLI. All SLIs are extracted from German Research Centre (GFZ) RSL database where SLIVISU software was used to define a proper regional subset. SLIVISU, visualization framework for
analysis and evaluation of model simulation was developed at GFZ (Unger et al., 2012) and is accessing the RSL database containing different compilations of records where, for this study, we use indicators for the Hudson Bay as a prominent region of GIA in Canada (Figs. 2(a) (b)), which were compiled by Art Dyke.

### 2.1    Creating height PDF

Each indicator has assigned height information based on author's studies in the SLI database compiled by Art Dyke (unpubl.).
However, the assigned height presents only the samples elevation relative to a datum, usually present sea level. We do not follow here further information given in the datebase to derive a relative sea-level range. In contrast, we consider the elevation as stating, that the depth of the sample, relative to RSL, is based on the conditions under which it was deposited. Following the work of Hibbert et al. (2016), we map the indicator's depth to RSL by an appropriate transfer function. The term transfer



function is not related to a specific numerical method, but the indicator's depth which is transformed in regard to modern depth information, based on the sample's type.

From the sample's material specified in the database, genus and species level are identified and the information from the Ocean Biogeographical Information System (OBIS) is extracted, listing the actual water depth at which the sample was found. The comprehensive OBIS database contains 45 million observations of nearly 120,000 marine species. The quality of the data

is undisputed coming from various institutes around the globe and passing quality control before publication (OBIS, 2017).

For this feasibility study we consider types of shells which were dated in the Hudson Bay and, thus, are of specific interest for GIA models. The Hudson Bay is located in north-eastern Canada (Fig. 2 (c)) and is in close proximity to the former glaciation center of the Laurentide ice sheet (Fig. 3). Hence, this area is widely explored by different studies regarding relative sea-level change (e.g., Dyke and Peltier, 2000; Fang and Hager, 2002). Here, we took 4 different types of shells into consideration;

Hiatella arctica, Macoma balthica, Portlandia arctica, and Mytilllus edulis. The shells' depth range is derived from the OBIS database, giving us necessary information for the transfer function with their association to the GIA signal. Figures 2(b) and 2(c) summarize the temporal and spatial distribution of the SLIs and Fig. 2(a) shows the heights distribution condensing them into a sea-level curve. The relation to GIA at last glacial maximum is shown in Fig. 3 (see supplement for the considered attributes from the database).

Extracting the depth distribution for each shell species from OBIS database, we find a good representation through a gamma distribution (Fig. 4). This kind of distribution is motivated as it is suitable for hydrological frequency analysis (Yue et al., 2001; Bobee and Ashkar, 1991; Clarke, 1980). If we observe OBIS data as independent data points $x = [x_1,...,x_n]$ from the same density, and we say that $x$ has a gamma distribution, with the PDF

$$f(x) = \frac{1}{a\Gamma(b)} \left(\frac{x}{a}\right)^{b-1} e^{-x/a}, \tag{1}$$

where $a>0$ and $b>0$ are shape and inverse scale parameters, respectively, meaning that this distribution is a uni-variate gamma distribution with two parameters. The two parameters are estimated via the maximum likelihood method (e.g., Moran, 1969):

$$s = \ln\left(\frac{1}{N}\sum_{i=1}^{N} x_i\right) - \frac{1}{N}\sum_{i=1}^{N} \ln(x_i) \tag{2}$$

Then, the estimator of $a$ is approximated by

$$\hat{a} \simeq \frac{3 - s + \sqrt{(s-3)^2 + 24s}}{12s} \tag{3}$$

and $b$ follows as

$$\hat{b} = \frac{a}{N}. \tag{4}$$

The gamma distribution is "shifted" by the observational height value of the specific SLI, and by this is representing the transfer function ($x \rightarrow x - \text{SLI}$).





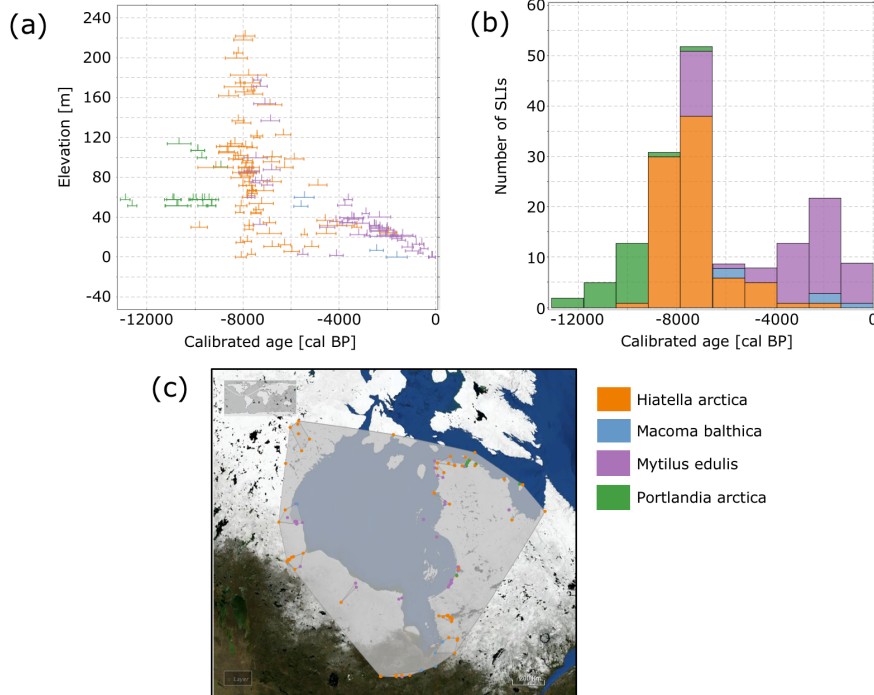

**Figure 2.** Data output from SLIVISU: (a) histogram depicting occurrence of selected SLIs in various age periods in the Hudson Bay. (b) Sea-level curve of the selected SLIs in the Hudson Bay. (c) Map of Hudson Bay with the selected SLIs retrieved from SLIVISU. Colors distinguish selected shells as shown in the color bar on the right.

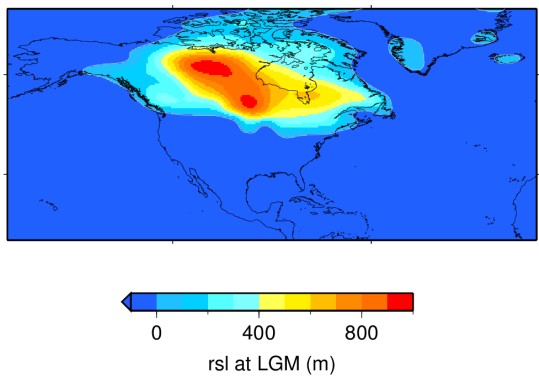

**Figure 3.** Relative sea level (RSL) at the Last Glacial Maximum, 26.5 ky BP, in the region of Hudson Bay.



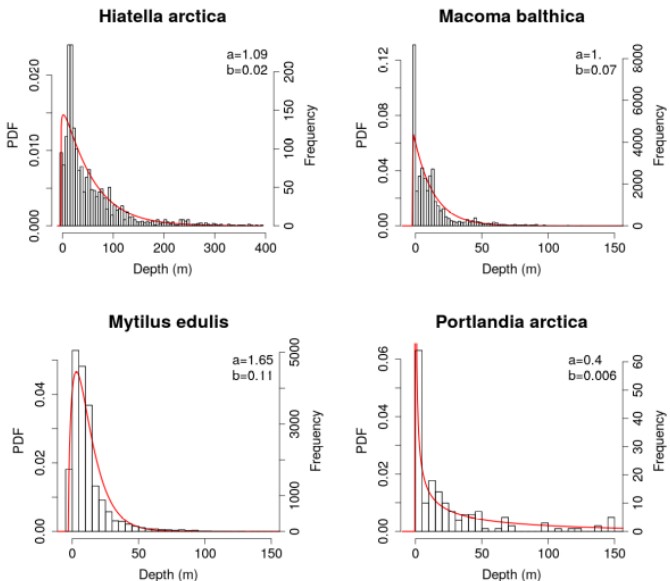

**Figure 4.** Depth distribution of selected shell types generated from OBIS (2017) and fit of respective gamma distribution (red curve).

In addition to the indicative meaning, each sample's depth is attributed to additional measurement errors which we have to account for. We assume them to be normally distributed, i.e.,

$$g(x|x_m, \sigma) = \frac{1}{\sqrt{2\pi\sigma^2}} e^{\frac{-(x-x_m)^2}{2\sigma^2}}, \tag{5}$$

where

$$\sigma = \sqrt{(\sigma_{\text{OBIS}})^2 + (\sigma_{\text{SLI}})^2} \tag{6}$$

sums up the uncertainties derived from the leveling of the OBIS data, $\sigma_{\text{OBIS}}$, and those of the SLI, $\sigma_{\text{SLI}}$. Assuming the observational errors and the depth distribution are represented as independent random variables, their combination is represented by a convolution of the two distributions

$$ph(x) = (f * g)(x) = \int\limits_{-\infty}^{\infty} f(y)\, g(x-y)\, dy. \tag{7}$$

For its calculation we apply the Fourier transformation.

## 2.2 Creating age PDF

Radiocarbon dating assumes a constant concentration of $^{14}C/^{12}C$ although this ratio is susceptible to the changes in the production of $^{14}C$ in the upper atmosphere. Therefore, a further age calibration is needed. This is done with OxCal (see Fig. 5), a





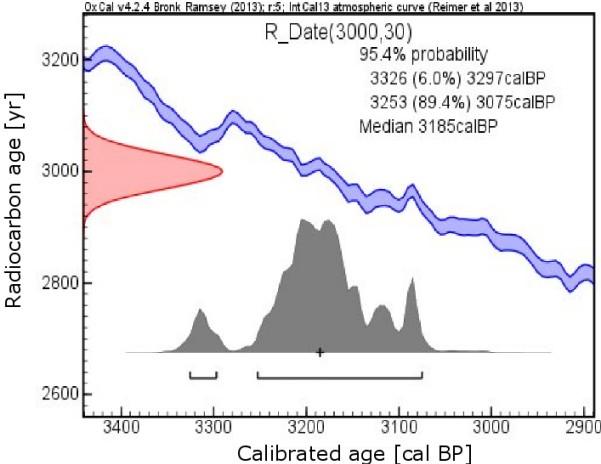

**Figure 5.** Example of calibration of OxCal output with measured radiocarbon determination of 3000±30 BP (Ramsey, 2017). Red curve on the left presents the age of the sample with considered Gaussian distribution, blue curves indicate the time range from the tree-ring with based calibration curve. Grey curve presents the posterior probability distribution of the calibrated age.

program designed for the analysis of chronological information (Ramsay, 2017). OxCal was developed at University of Oxford and it provides the use of most current calibration curves with the possibility to apply reservoir corrections based on sample's habitat.

For the time range of considered SLIs, the IntCal13 curve (Reimer et al., 2013) with the default resolution of 5 years, is applied. Therefore, no interpolation or binning is needed. The posterior probability distribution is calculated by the calibration

program as the age PDF $pa^{\text{SLI}}(t)$. In addition, we have to account for the fact that Portlandia arctica and Macoma balthica are deposit feeders, meaning, they absorb bicarbonate from the rocks they live on, unlike suspension feeders, causing them to appear older by up to 2000 [14]C years (England et al., 2013). Therefore, if such SLIs are located in regions where deposits are calcareous, their [14]C concentration is affected by an unknown fraction of practically [14]C-free carbonate from million years old rock, making them unreliable for chronological reconstructions (England et al., 2013).

In Fig. 6, we compared the locations of SLIs (satellite image on the left) with calcareous regions (bright blue regions on the geological map right) and disregarded 4 indicators belonging to Macoma balthica type, located in the most southern part of Hudson bay, James Bay. Other points are taken into account since they are located on the Precambrian shield (red region on the right).

## 2.3  Creating joint PDF

The two PDFs representing the uncertainties in age and height are used to create joint probability densities. Referring to the theory from Tarantola (2005), the joint PDF is equal to the product of two marginal probabilities. With this assumption we can



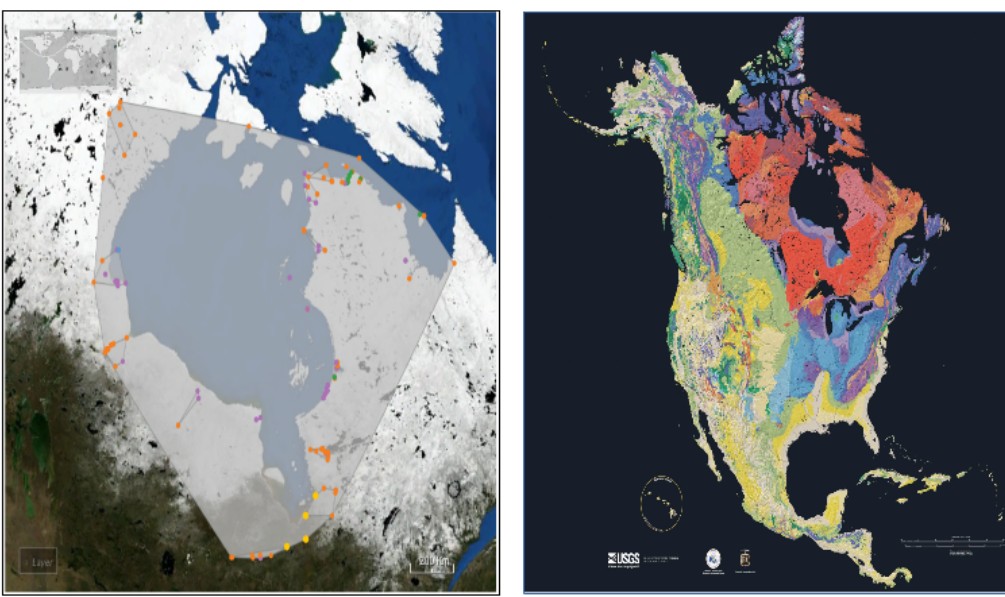

**Figure 6.** Image on the left shows the location of all selected indicators from Hudson bay region as retrieved from SLIVISU, where yellow dots are representing Macoma balthica SLIs that are excluded from the study. Other colors apply to shell types as in Fig. 2. Image on the right represents geology of North America, where light blue color depicts potentially calcareous regions and red the Precambrian Canadian shield (image courtesy of USGS).

represent the joint PDF as

$$f(t, h) = pa(t)\, ph(h)\,. \tag{8}$$

Figure 7 shows a typical example for Mytilus edulis, where we distinguish the confidence intervals of 68%, 95%, and 99%. The asymmetries with respect to height and age are clearly visible.

### 2.4 Evaluation of the fit to the model prediction

5 For the evaluation of numerical models of reconstructing former sea level, $h_{\mathrm{RSL}}$, we continue by considering the conditional probability density based on the definition of Tarantola (2005) on page 18. This probability density is a special case of conjunction of probability distributions (in our case height and age probability). In other words, we want to 'condition' the joint probability (9). Thus, if we say that the joint PDF represents application of $ph$ to $pa$, our condition is to get the values where $ph = ph(t)$. For the time interval covered by the confidence of $pa$, we assume $h_{\mathrm{RSL}}$ to be linear,

10 $$h_m^{\mathrm{RSL}}(t, \Omega) = a_m + b_m\, t\,, \tag{9}$$

here $a_m$ and $b_m$ are predicted height and uplift velocity, respectively, at the location $\Omega$.





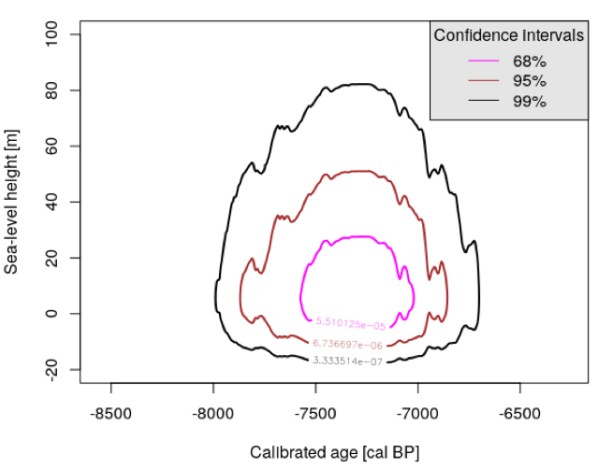

**Figure 7.** Joint probability density of Mytilus edulis presented as a 2d contour plot.

Then, the conditional probability can be shown to follow

$$F_{h,t/m} = \int pa^{\mathrm{SLI}}(t)\, ph^{\mathrm{SLI}}(h_m(t))\, dt \,. \tag{10}$$

After calculating the joint PDF (8) for each selected SLI, we continue with the conditional probability (10) for a whole set of SLIs for each model of the ensemble we want to validate (model parameters are presented in Tab. 1).

Assuming that the conditional probabilities of the individual SLIs, $P_i$, are independent, the joint probability results as their

5 intersection

$$P = \bigcap_{i=1}^{N_{\mathrm{data}}} P_i = \prod_{i=1}^{N_{\mathrm{data}}} P_i \,, \tag{11}$$

which is computed as the product of the respective PDFs. For computational convenience, we consider the logarithm of the distribution,

$$\ln \mathcal{L} = \frac{1}{N_{\mathrm{data}}} \sum_{i=1}^{N_{\mathrm{data}}} \ln(P_{\mathrm{sli}_i}) \,. \tag{12}$$

10 which is then divided by the total number of samples $N_{\mathrm{data}}$. For the conditional probability, we again assume different indicators to be independent, which results in the intersection to be calculated as the sum of the samples' conditional probabilities in the logarithm scale.





## 3 Results

For this study we consider an ensemble of sea-level reconstructions, which was generated in the German Climate Modeling initiative PalMod, and represents the variability in $h_{\mathrm{RSL}}$ due to variations in the Earth's structure with respect to lithosphere thickness, upper- and lower-mantle viscosity (Table 1). The model predictions were calculated with VILMA, Viscoelastic Lithosphere and Mantle model, for modeling of global deformations and gravity changes (Martinec, 2000). We analyzed a

model ensemble of GIA reconstructions containing 140 different members. For all models, the glaciation history ICE6G_C (Peltier et al., 2015) was applied.

**Table 1.** List of Earth-structure variability parameters on which the model ensemble is based.

| Parameter | Values |
| --- | --- |
| Lithosphere thickness [km] | 60, 80, 100, 120 |
| Upper-mantle viscosity [$10^{21}$ Pa s] | 0.1, 0.2, 0.5, 0.8, 1 |
| Lower-mantle viscosity [$10^{21}$ Pa s] | 1, 2, 5, 8, 10, 20, 50 |

Each ensemble member provides predicted height, $a_m$, and slope, $b_m$, at the location and age, we consider here the median of the respective SLIs. First, we reduce the integration range of $ph$ that is in $pa$ range by determining minimum and maximum of the range

$$h_{\min} = a_m(t_{\min} - t_m) + b_m$$
$$h_{\max} = a_m(t_{\max} - t_m) + b_m$$

(13)

where $t_{\min}$, $t_{\max}$, and $t_m$ present minimum, maximum of the 99.5 confidence region of $pa$, and the median of calibrated age, respectively. Next, we re-sample $ph$ according to the model dependent range $(h_{\min}, h_{\max})$, in order to solve (11) numerically to estimate the likelihood of one indicator represented by one ensemble member.

After examination of the values, we have determined the 'best' fit for the model with lithosphere thickness of 60 km, upper-

mantle viscosity of $5 \times 10^{20}$ Pa s and lower-mantle viscosity of $5 \times 10^{21}$ Pa s. Next, we use this model fit as a reference model and compare it to different model predictions using Bayesian model comparison. As for this study, we do not derive a posterior probability, we base our model fit on Bayes factors (e.g., Kass and Raftery, 1995), which for two hypothesis we can write as:

$$K = \ln(P_1/P_2) = \ln P_1 - \ln P_2 \qquad (14)$$

For $P_1$ we assume the model fit with the highest value. In Fig. 8, the results are presented graphically as a 3D scatter plot,

where models with better fit have more intense colors. As expected from the wide probability intervals, we get a large scatter of acceptable models fitting the considered set of indicators, but with a preference of models with upper-mantle viscosities of slightly less than $1 \times 10^{21}$ Pa s.



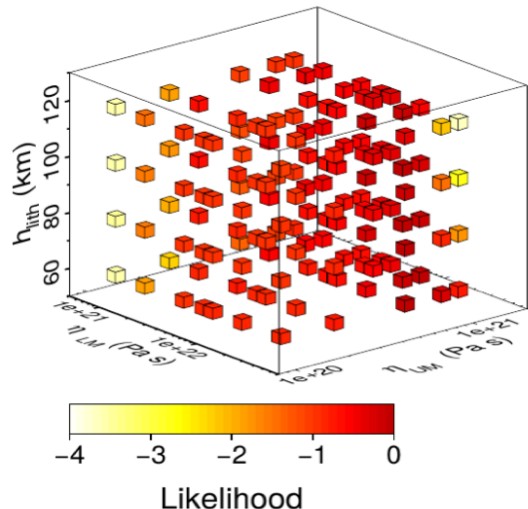

**Figure 8.** 3D presentation of model ensemble with 140 members varying in lithosphere thickness ($h_{\text{lith}}$), upper- and lower-mantle viscosity ($\eta_{\text{UM}}, \eta_{LM}$). Color scale indicates likelihood to consider set of shallow-water shells of Hudson bay region.

Isolating slices of the considered lithospehere thicknesses (Tab. 1) are displayed in Fig. 9. The contour patterns are showing best fits in the same region, which corresponds to region values of $5 - 8 \times 10^{20}$ Pa s for upper-mantle viscosity and values of $2 - 5 \times 10^{22}$ Pa s for lower-mantle viscosity, with slightly better fits for models with lithosphere thickness of 60 and 80 km.

Wolf et al. (2006) did a thorough comparison of studies estimating mantle viscosity for Hudson Bay region and they are summed up in the Table 2, including their results, the study from Zhao (2013), few studies with global estimates and the current

5  one.

Based on the values from previous studies we can conclude that this method overestimates the values of lower-mantle viscosity. While majority of studies are estimating lithosphere thickness at 120 km we identify a slight preference for lower thicknesses. But, it is important to mention that this findings are only meant to explain the method and not actually to constrain models. Here we restricted ourselves to a low quality indicator type. So, discussing other types of samples like upper limits

10  would better constrain the parameter space.





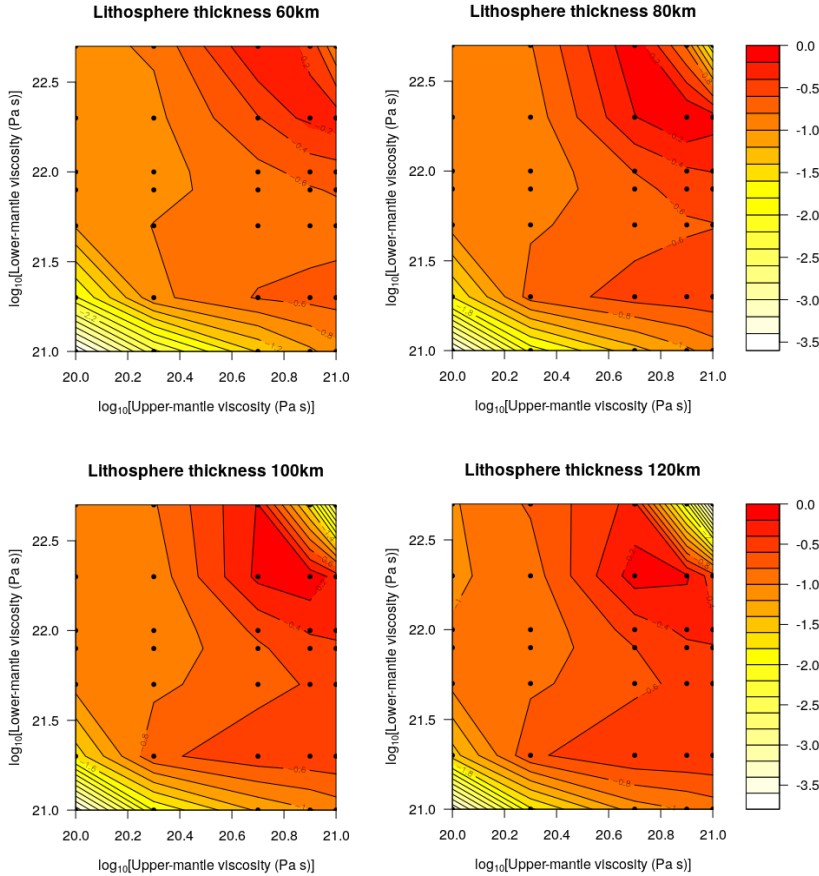

**Figure 9.** Model fits as function of upper- and lower-mantle viscosities for considered lithosphere thicknesses.





**Table 2.** List of different estimates of mantle viscosity, $\eta_{UM}$, $\eta_{LM}$, based on Wolf et al. (2006), from the Hudson Bay region complemented by more recent studies and global estimates, the latter, noted by *.

| Publication | Viscosities [$10^{21}$ Pa s] | |
| --- | --- | --- |
| | Upper-mantle | Lower-mantle |
| Nakada (1983) | 0.05–0.75 | 100 |
| Peltier and Andrews (1983) | 1 | 1–3 |
| Nakada and Lambeck (1991) | 4–6 | 20–50 |
| Mitrovica and Peltier (1992) | 1 | 1–3 |
| Han and Wahr (1995) | 1 | 50 |
| Mitrovica and Peltier (1995) | 0.5 | 0.5–3 |
| Lambeck (1998)* | 0.3 | 10 |
| Cianetti et al. (2002) | 1 | 2 |
| Mitrovica and Forte (2002) | 0.39–0.43 | 6.5–11 |
| Mitrovica and Forte (2004) | 0.5 | 1 |
| Wolf et al. (2006) | 0.32 | 16 |
| Zhao (2013) | 0.37 | 1.9 |
| Peltier et al. (2015)* | 0.5 | 3 |
| Lambeck et al. (2017)* | 0.35–0.75 | 8–28 |
| this study | 0.5–0.8 | 20-50 |





## 4 Conclusions

Over the past decades, various glacial isostatic adjustment (GIA) models were developed, and more recently, also considering a 3D Earth structure (e.g., Wu et al., 1998; Martinec, 2000; Forno et al., 2005; Latychev et al., 2005; Whitehouse et al., 2006; Klemann et al., 2008; van der Wal et al., 2013). Interpretation of vast geological data for purpose of model validation and numerical constrains often remains subjective. Here, we follow the statistical model proposed by Hibbert et al. (2016). For

each sea-level indicator (SLI), we calculate probability density functions based on the observational error distribution and the indicative meaning of the considered species. The latter is derived from the OBIS database for the water depth distribution at which it was found to live. In addition, we consider the calibration statistics of the radiocarbon dated samples and combine them to a non-normally distributed joint probability.

  In this study, as an application of the method, we considered a total number of 156 SLIs covering the Hudson Bay region

for the period since 12,000 years BP. Based on the glaciation history ICE6G_C, we determined the best fitting Earth models to be represented by upper- and lower mantle viscosities of $5 - 8 \times 10^{20}$ Pa s and $2 - 5 \times 10^{22}$ Pa s, respectively. In comparison to other inferences of viscosity structures, our findings deviate slightly which might be due to our restriction to shallow water shells. But, as mentioned before, this study focuses on the explication of the method. Therefore this result should be taken as preliminary. For applying the method, further types of SLIs should be investigated, and further corrections to the indicators like

tides should be included. This method proved to be suitable for constraining model ensemble based sea-level change caused by GIA. It will allow us to exploit a large number of SLIs, that are usually disregarded due to low-quality, in estimations of RSL change in regions and on time scales where indicators of higher quality are not available.

*Data availability.* see supplement which will be provided together with the submisison

*Competing interests.* We see no competing interest at present

*Acknowledgements.* The research was funded by German Federal Ministry of Education and Research (BMBF) as Research for Sustainability initiative (FONA); www.fona.de through the PalMod project. Furthemore, this study benefited a lot from discussions inside PALSE 2. We thank Art Dyke for providing his database of sea-level indicators.



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
