# Peer review of "A statistical method to validate reconstructions of late-glacial relative sea level – Application to shallow water shells rated as low-grade sea-level indicators"

_Climate of the Past, 2018_

## Short Comment (SC1) · Milena Latinović et al. · 19 Jun 2018

I appreciate the statistical approach proposed by Latinovic et al – it offers a quantitative solution to the vertical data of a sea-level index point in particular to its uncertainty which is often estimated through an informed decision only. Assessing to the highest precision possible the water-depth range in which biological sea-level indicators live is a significant step forward. Here, I wish to comment on some sentences which are, I believe, misleading, possibly due to some hard to read idiomatic expressions. I feel the text on sea-level indicators (page 2 line 9ff) is confusing: I wouldn't know any reliable

biological indicator that form at the shoreline. These live at different water depths, i.e. in front of the shoreline or, if terrestrial, behind the shoreline. The indicator's height at time of sampling does not provide the 'indicative meaning' but just its position as encountered today. 'Indicative meaning' (van de Plassche 1986) is the indicator's living range with respect to the corresponding shoreline at the time when the species was alive. It comprises two parameters: the 'reference water level' (i.e. the elevation of the indicator as identified through a modern analogue) and the 'indicative range' (i.e. the vertical range over which the indicator's modern analogue exits) both contributing to the vertical uncertainty of a sea-level index point (e.g. Shennan et al., 2015). I believe the authors are after 'indicative range', not after 'indicative meaning'. They wish to improve the vertical data point in terms of its precision using the Bayes' theorem. This is very useful because it reduces the uncertainty of the sea-level prediction. Also, may I suggest to follow Shennan (2015) and replace the term 'relative sea level' with 'sea level' because the sea level is the difference between the geoid and the solid earth surface relative to the centre of the earth (Mitrovica and Milne, 2003). In fact, there is no difference between the sea level as defined by the geological community and the mathematical notation of the geophysicists (Shennan, 2015). This would facilitate reading (e.g. "...the depth of the sample, relative to RSL..."; page 3, line 17). In addition, may I suggest to replace the term 'transfer function' with other words (e.g. 'conversion by analogy' or similar). 'Transfer function' is used since the 90s in numerous publication (see Barlow et al., 2013 and references herein). The function describes the statistical relationship between the modern vertical distribution of diatoms or foraminifera and their fossil counterparts in order to establish the indicative meaning of a sea-level index point and its vertical uncertainty. The text (page 4, line 1) indicates that authors mean 'indicative range'.

Shennan, I., 2015. Handbook of sea-level research: framing research questions, in: Shennan, I., Long, A.J. and Horton, B.P. (eds)., Handbook of sea-level research, Wiley, pp.3-25. Mitrovica, J.X. and Milne, G., 2003. On post-glacial sea level: I. General theory. Geophys. J. Int. 154, 253–267. Barlow, N., Shennan, I., Long, A.J., Gehrels,

W.R., Saher, M.H., Woodroffe, S.A. and Hillier, C., 2013. Salt marshes as late Holocene tide gauges. Global and Planetary Change 106, 90-110.

---

## Short Comment (SC2) · 21 Jun 2018

Dear Barbara Mauz,

Thank you for the interest in our manuscript, helpful suggestions and for your contribution to the discussion forum of this manuscript.

We agree with your first comment that the text on page 2, line 9 is misleading. We replace the first two sentences by: 'In the contrast to recent measurements by, e.g. tide gauges, sea level indicators are geological samples which were deposited usually

below or above sea level at that time.'

Indeed, the indicator's height does not provide the 'indicative meaning', but it contains indicative meaning regarding the samples deposition or depositional conditions with respect to the former sea level (van De Plassche 1986), and we agree that is more precise to use the term 'indicative range' for the vertical range of the sample in relation to the modern analogue.

We will keep the term 'relative sea level' to distinguish this measure from the absolute sea level which is measured against a fixed reference frame like the center of mass. However, you are right to point out the redundancy in the sentence on page 3, line 17. We will replace it by 'relative to the local sea level'.

We understand transfer function as mathematical term relating the input to the corresponding output by applying additional operations. Therefore, we keep this terminology where we map the indicator's depth to relative sea-level based on present day depth.

Once again, thank you for your useful suggestions, which we will consider in the revised version of the manuscript.

Kind regards,

Milena Latinović and Volker Klemann

van de Plassche, O., ed.: Sea-level Research: A Manual for the Collection and Evaluation of Data, Norwich, Geo Books, Norwich, https://doi.org/10.1007/978-94-009-4215-8, 1986.

---

## Referee Comment (RC1) · L. Tarasov (Referee) · 13 Aug 2018

Review of : A statistical method to validate reconstructions of late-glacial relative sea level – Application to shallow water shells rated as low-grade sea-level indicators

It is great to see a study that considers how to rigorously define conditional probability distributions for RSL for paleo contexts. However, the current submission has a major flaw. The SLI residuals are not independent and this must be explicitly accounted for. The current formulation explicitly assumes independence but then contradicts this with a 1/N normalization. The consequence of SLI dependence is clear, for instance, when considering the whole Dyke RSL database for North America. The spatial-temporal density of RSL datapoints varies greatly with resultant variations in datapoint redundancy.  Without taking this density variation explicitly into account, use of your scoring scheme for say deglacial ice sheet model calibration will give results with model-data fits biased to where datapoints density is highest, even if the sectors where this occurs represent just a small area fraction of the LGM North American ice complex. Until this is addressed, the statistical method is invalid.

I should also note that this flaw might have been avoided with a more careful consideration of the existing litterature (which is not evident in the reference list), eg Briggs and Tarasov, 2013 and Love et al, 2016.

I do not understand the choice of journal.  This submission would seem to me much more appropriate in GMD especially since the novelty here isn't the theory (this is standard Bayesian and probability theory) but the actual implementation. The first line in the abstract also delineates this as a methodology paper: "In this study, we propose a statistical method to validate sea-level reconstructions using geological records known as sea-level indicators (SLIs)." Futhermore, the paper focus is on the method with the viscosity results only provided as an example : "findings are only meant to explain the method and not actually to constrain models."

The paper would also strongly benefit from more concrete details on implementation (probably best included in the supplement) to enable others to do so (especially since the software toolbox is not being made available).

Submission to GMD though requires provision of necessary code/software. This then raises an inequity between the two journals, submit to CPD and avoid the need to provide required code.... I'll defer the appropriate journal choice to the Chief Editor who should have a clearer sense of journal scope. I would like to see a statement from the editor clarifying how to resolve the scope intersection between GMD and CP with respect to software availability.

I would also like to see explicit consideration of tidal range and wave impacts, especially given the significant tides in Hudson Bay along with the well-known "storm-beach" displacement of SLIs.

Once these issues (and the points below) are addressed, I would see this submission as worthy of publication in GMD (or CP if justified by the chief editor).

```
**specific comments**

For this conceptual study, we restrict ourselves to one type of
indicators, shallow10 water shells, which are usually considered as
low-grade samples giving only a lower limit of former sea level, as
the depth range in which they live spreads over several tens of
meters, and does not follow a normal distribution
**This statement is too sweeping. Eg Dick Peltier and myself treat**
**certain inter-tidal species (eg Myt. Ed.) as providing more than**
**just 1-way bounding.**

The shells' depth range is derived from the OBIS
database,
**You need to make clear whether the database only includes shells**
**that were found in living position as well as whether the shells**
**were living or not.**

In addition to the indicative meaning, each sample's depth is
attributed to additional measurement errors which we have to account
for. We assume them to be normally distributed, i.e.,
**It should be stated whether all considered SLIs were found in a living positio**
n.
**If not, how are the additional uncertainties addressed?**

sums up the uncertainties derived from the leveling 5 of the OBIS data,
sigma_OBIS, and those of the SLI, sigma_SLI.
**I'm confused. Doesn't the gamma function account for sigma_SLI?**

For the time range of considered SLIs, the IntCal13 curve
**Why wasn't the marine calibration curve used? Furthermore there**
**needs to be accounting of Reservoir age uncertainties (and reservoir**
**age itself if you are using the IntCal curve).  The text should also**
**briefly describe reservoir ages uncertainties (given their**
**non-trivial space/time variations).**

**Fig 3. LGM RSL is kind of meaningless since all the SLIs are only present**
**after local deglaciation. Better to show eg 8 ka RSL around when most of**
**the critical Hudson Bay dates are available.**

eq 9: here am and bm are predicted height and uplift velocity
**a_m is the predicted height at t=0 only**

lithospehere -> lithosphere

Fig 9
**please use a higher contrast colour scheme to make this easier to read**

Notation: equations 10-12
**Use consistent notation. Eg eq 11 use P_i for conditional probability**
**but eq 10 uses F_{h,t/m}. Best would be to use standard statistical**
**notation for conditional probability, eg (h|x) for h conditioned on x.**

eq 10
```

```
**How is this implemented? And how is pa(t) retrieved from oxcal?**
**oxcal is a complex enough application that a bit of guidance here**
**would help others with their own implementation.**

Assuming that the conditional probabilities of the individual SLIs,
P_i, are independent, the joint probalility
eq 12

**the 1/N_data normalization in eq 12 breaks the stated assumption of**
**independent conditional probabilities.  The likelihood is the joint**
**conditional probability given by P in eq 11.  ln(L) would just be**
**SUM(ln(P_sli_i) if the residuals were truly independent. Anyway,**
**there is no basis to assume all the SLI residuals are**
**independent.**
```

---

## Referee Comment (RC2) · Anonymous Referee #2 · 24 Aug 2018

**A statistical method to validate reconstructions of late-glacial relative sea level – Application to shallow water shells rated as low-grade sea-level indicators**

Milena Latinović1,2, Volker Klemann1, Christopher Irrgang1, Meike Bagge1, Sebastian Specht1,3, and Maik Thomas1,2

1GFZ German Research Centre for Geosciences, Potsdam, Germany
 2Free University, Berlin, Germany
 3University of Potsdam, Potsdam, Germany

Correspondence: Milena Latinović (milena.latinovic@gfz-potsdam.de)

Abstract. Observations of sea-level variations allow the validation of numerical models used to reconstruct past and predict future sea-level change. Sea-level indicators (SLIs) are used as the main source for deriving relative sea-level (RSL) variations during previous epochs for which tide gauge and satellite measurements were not yet available. However, the leveling of an

- 5 SLI relative to present sea level does not provide a direct measure of former RSL, but only an indication according to the conditions under which the sample was deposited. This information depends on the sample type and on its environment and has to be mapped to RSL by an appropriate transfer function. The respective data has to be extracted by an objective procedure from primary information usually provided in geological or palaeontological literature of different primary focus, quality, and detailedness. In addition to the height information, also the precision of dating varies between different indicators and, in case
- 10 of radiocarbon-dated material, a further calibration of the dated age has to be applied.

[revised manuscript text omitted]

---

## Author Comment (AC1) · 19 Sep 2018

**General comments**

1. It is great to see a study that considers how to rigorously define conditional probability distributions for RSL for paleo contexts. However, the current submission has a major flaw. The SLI residuals are not independent and this must be explicitly accounted for. The current formulation explicitly assumes independence but then contradicts this with a 1/N normalization. The consequence of SLI dependence is clear, for instance, when considering the whole Dyke RSL database for North America. The spatial−temporal density of RSL datapoints varies greatly with resultant variations in datapoint redundancy. Without taking this density variation explicitly into account, use of your scoring scheme for say deglacial ice sheet model calibration will give results with model−data fits biased to where datapoints density is highest, even if the sectors where this occurs represent just a small area fraction of the LGM North American ice complex. Until this is addressed, the statistical method is invalid.

I should also note that this flaw might have been avoided with a more careful consideration of the existing literature (which is not evident in the reference list), eg Briggs and Tarasov, 2013 and Love et al, 2016.

We fully agree that we should consider the spatial-temporal distribution of RSL datapoints. Briggs and Tarasov (2103) as well as Love et al. (2016) applied a spatial weighting algorithm to already aggregated curves in order to consider the clustering of curves in specific regions. In this study, the individual SLIs are analyzed independently. Accordingly, we apply the redundancy weighting method proposed in Caron et al. (2017) where the cross correlations of the SLIs with respect to the considered model ensemble are taken into account. Therein, for each SLI a redundancy weight $w_i = \dfrac{K}{\sum\limits_{j=1}^{N_{data}} \rho_{ij}}$ is defined.

Here, $K$ is a normalization constant so that $\sum\limits_{i} \dfrac{w_i}{N_{data}} = 1$ , $N_{data}$ is the total number of data. The Pearson correlation coefficient between the ensembles of predictions $i$ and $j$ is represented as $\rho_{ij} = \dfrac{cov(i,j)}{\sigma_i \sigma_j}$ where $cov(i,j)$ is the covariance between two SLIs and $\sigma_i$, $\sigma_j$ are the standard deviations of the two SLIs. The redundancy weights are calculated for each SLI and are considered as prefactors in Eq. 13, that now reads:

$$\ln L = \frac{1}{N_{data}} \sum_{i=1}^{N_{data}} w_i \ln\left(P_{sli_i}\right)$$

[Figure]

**Figure x.** Redundancy weights for each selected SLI. The color of the individual circle denotes considered weight, the shade of the time around denotes the calibrated age of the SLI. An overlap of SLIs could not be avoided. Dashed line separates Hudson-Strait indicators from those of the Hudson Bay.

In addition to the above procedure explanation of calculating the weights, we also included this figure into the manuscript

2. I do not understand the choice of journal. This submission would seem to me much more appropriate in GMD especially since the novelty here isn't the theory (this is standard Bayesian and probability theory) but the actual implementation. The first line in the abstract

also delineates this as a methodology paper: "In this study, we propose a statistical method to validate sea−level reconstructions using geological records known as sea−level indicators (SLIs)." Furthermore, the paper focus is on the method with the viscosity results only provided as an example : "findings are only meant to explain the method and not actually to constrain models."

The paper would also strongly benefit from more concrete details on implementation (probably best included in the supplement) to enable others to do so (especially since the software toolbox is not being made available).

Submission to GMD though requires provision of necessary code/software. This then raises an inequity between the two journals, submit to CPD and avoid the need to provide required code.... I'll defer the appropriate journal choice to the Chief Editor who should have a clearer sense of journal scope. I would like to see a statement from the editor clarifying how to resolve the scope intersection between GMD and CP with respect to software availability.

We did not consider GMD journal for submission since, as you mentioned, it requires a software, and we are not software developers. Providing the code that is user friendly would require more time. But we will, of course, respect editor's decision. We will include in the supplement a more detailed explanation of the algorithms considered and also an example of calibration with OxCal and how we extracted *pa(t).*

3. I would also like to see explicit consideration of tidal range and wave impacts, especially given the significant tides in Hudson Bay along with the well−known "storm−beach" displacement of SLIs. Once these issues (and the points below) are addressed, I would see this submission as worthy of publication in GMD (or CP if justified by the chief editor).

The tidal range for the Hudson Bay at present varies between 0 and 4 m (Webb 2014), and produces accordingly a small offset, which we decided to neglect for this study. In the outlook, we discussed the consideration of tidal ranges. In the majority of the studies from which we obtained the data (reference in the supplement of the manuscript), we could not find evidence that samples were found on the "storm-beach", apart from the local correlation suggested by the primary investigator. Therefore we did not consider the displacement of SLIs due to this effect in the study. But, since we established that Allard & Trembley (1983) related the samples from their study to "storm-beaches", as they found SLIs of 650 yr at an elevation of 4 m (Manitounuk islands), we will consider displacement in future studies. For indicators related to shore line, and not explicitly defined as picked from their living position (Klemann & Wolf, 2007), we will consider possible shift of 4 m (assumed storm beach height) .

We included this answer to the manuscript.

**Specific comments**

4. "For this conceptional study, we restrict ourselves to one type of indicators, shallow water shells, which are usually considered as low−grade samples giving only a lower limit of former sea level, as the depth range in which they live spreads over several tens of meters, and does not follow a normal distribution"
**This statement is too sweeping. Eg Dick Peltier and myself treat certain inter−tidal species (eg Myt. Ed.) as providing more than just 1−way bounding.**

Already visible from figure 4, Mytilus Edulis' living range is extending into the inter-tidal, meaning the lower limit of the depth range is smaller than 0. We merely wanted to indicate that only one of the boundaries is exactly specified close to 0 m depth.

5. "The shells' depth range is derived from the OBIS database,"
**You need to make clear whether the database only includes shells that were found in living position as well as whether the shells were living or not.**

The database does not explicitly say if the samples were found in the living position, but the intention of the database is to discuss environmental conditions of species, which led us to assume that they were found in the living position. Hibbert at al (2016) uses OBIS database for living position of corals and we followed this approach.

6. "In addition to the indicative meaning, each sample's depth is attributed to additional measurement errors which we have to account for. We assume them to be normally distributed, i.e.,"
**It should be stated whether all considered SLIs were found in a living position. If not, how are the additional uncertainties addressed?**

If considered SLIs are not found in a living position, our method is invalid, because we based our calculations on the specific location of each sample. Of course, it is tricky to ensure this information from the primary literature, so we decided to rely on this assumption.

7. "sums up the uncertainties derived from the leveling of the OBIS data, sigma_OBIS, and those of the SLI, sigma_SLI."
**I'm confused. Doesn't the gamma function account for sigma_SLI?**

Gamma function is only indicating depth distribution and not accounting for $\sigma_{SLI}$. Observational errors due to the leveling of the the depth in OBIS data and elevation of the SLIs are assumed as 1 m and 5 m, respectively.

8. "For the time range of considered SLIs, the IntCal13 curve"

**Why wasn't the marine calibration curve used? Furthermore there needs to be accounting of Reservoir age uncertainties (and reservoir age itself if you are using the IntCal curve). The text should also briefly describe reservoir ages uncertainties (given their non−trivial space/time variations).**

From the personal correspondence with Art Dyke we found out that, while gathering data for the database, he did Marine Reservoir age correction for 440 years for those SLIs that were not already corrected in the primarily reported age. We therefore used IntCal13 atmospheric curve to avoid double correction. But, we do agree that it would be more correct to use the marine curve for this type of indicators. So, we first added back 440yr that Art Dyke accounted for, and then we applied marine curve (Marine.13) on the selected SLIs. Butzin et al. (2017) discuss spatial and temporal variability of the marine radiocarbon reservoir age during the last 50,000 years based on ocean circulation modelling. The authors did not focus on small regions like the Hudson Bay. Nevertheless their published model results (Butzin et al, 2017, data), show some variability. Therein, we find a decrease of reservoir age from about 700 years in the Hudson Strait to about 416 yr in the Hudson Bay for the last 12,000 yr. The time variability amounts to 50 yr for this time interval. In some parts near the W and SW shoreline of Hudson Bay, the basin correction reduces further to 200 yr what we do not consider in this study. In contrast,  we split our data into two regions, 'Hudson Bay' and 'Hudson Strait', in which we consider basin corrections for the considered marine shells of 416 ± 50 yr and 700 ± 50 yr, respectively. The reason for this deviation is the higher sea-ice concentrations in Hudson Strait than in the central Hudson Bay. As sea ice inhibits air-sea $^{14}CO_2$ exchange, this leads to lower surface water concentrations (=> higher $^{14}C$ ages) in the entry of Hudson Bay than in the central bay (pers. comm. Martin Butzin).

Figure 7, after applying marine curve, now looks like the following one:

[Figure]

**Figure 7**. Joint probability density of Mytilus edulis presented as a 2d contour plot.

This explanation is included in the manuscript along with updated Figure 7. and brief description of reservoir age uncertainties.

9. # Fig 3. LGM RSL is kind of meaningless since all the SLIs are only present after local deglaciation. Better to show eg 8 ka RSL around when most of the critical Hudson Bay dates are available.

We changed the Fig 3. and replaced as you suggested.

[Figure]

**Figure 3**. Relative sea level (RSL) at 8,000 years before present, in the region of Hudson Bay.

10. eq 9: here am and bm are predicted height and uplift velocity
**a_m is the predicted height at t=0 only**

Corrected Eq. 9:   $h_m^{RSL}(t,\Omega)=a_m+b_m(t-t_m)$   where $t_m$ is the median of calibrated age.

11. lithospehere −> lithosphere

Corrected

12. Fig 9 # please use a higher contrast colour scheme to make this easier to read

Corrected, Figure 9 is replaced with the following with recalculated fits.

[Figure]

**Figure 9**. Model fits as function of upper- and lower-mantle viscosities for considered lithosphere thicknesses

13. Notation: equations 10−12
Use consistent notation. Eg eq 11 use P_i for conditional probability but eq 10 uses F_{h,t/m}. Best would be to use standard statistical notation for conditional probability, eg (h|x) for h conditioned on x. eq 10

We changed the notation in the Eq. 10 that now reads as follows:

$$P_{h,t|m} = \int_{-\infty}^{\infty} pa^{SLI}(t)\, ph^{SLI}\left(hm^{RSL}(t)\right) dt$$

14. # How is this implemented? And how is pa(t) retrieved from oxcal?
**oxcal is a complex enough application that a bit of guidance here would help others with their own implementation.**

Based on your comment from the beginning, we included explanation about how we used OxCal to calibrate SLIs and how we retrieved *pa(t)* in the supplement.

15. Assuming that the conditional probabilities of the individual SLIs, P_i, are independent, the joint probalility eq 12
**the 1/N_data normalization in eq 12 breaks the stated assumption of independent conditional probabilities. The likelihood is the joint conditional probability given by P in eq 11. ln(L) would just be SUM(ln(P_sli_i)) if the residuals were truly independent. Anyway, there is no basis to assume all the SLI residuals are independent.**

As discussed in the beginning of the review, we agreed that the residuals are not independent by including the spatial-temporal weights. After re-calibration with marine curve and calculation of fits with weights, we got similar results as before; $5 - 8 \times 10^{20}$ Pa s for upper-mantle viscosity and values of $2 - 5 \times 10^{22}$ Pa s for lower-mantle viscosity with lithosphere thickness of 60 and 80 km.

**Literature**

Allard, M. and Trembley, G. (1983), La dynamique littorale des ıˆles Manitounuk durant l'Holoce`ne, Z. Geomorphol. suppl. 47, 61–95.

Briggs R. and Tarasov L.. Evaluating model-derived deglaciation chronologies for Antarctica. Quaternary Science Reviews, 2013. http://dx.doi.org/10.1016/j.quascirev.2012.11.021

Butzin, Martin; Köhler, Peter; Lohmann, Gerrit (2017): Marine radiocarbon reservoir ages for the past 50,000 years, links to model results in NetCDF format. PANGAEA, https://doi.org/10.1594/PANGAEA.876733, Supplement to: Butzin, M et al. (2017): Marine radiocarbon reservoir age simulations for the past 50,000 years. Geophysical Research Letters, 44(16), 8473-8480, https://doi.org/10.1002/2017GL074688

Caron, L., Ivins, E. R., Larour, E., Adhikari, S., Nilsson, J., and Blewitt, G.: GIA Model Statistics for GRACE Hydrology, Cryosphere, and Ocean Science, Geophys. Res. Lett., 45, 2203–2212, https://doi.org/10.1002/2017GL076644, 2018

Hibbert, F., Rohling, E. J., Dutton, A., Williams, F., Chutcharavan, P. M., Zhao, C., and Tamisiea, M. E.: Coral indicators of past sea-level change: A global repository of U-series dated benchmarks, Quat. Sci. Rev., 145, 1–56, 2016.

Klemann V., Wolf D. (2007) Using Fuzzy Logic for the Analysis of Sea-level Indicators with Respect to Glacial-isostatic Adjustment: An Application to the Richmond-Gulf Region, Hudson Bay. In: Wolf D., Fernández J. (eds) Deformation and Gravity Change: Indicators of Isostasy, Tectonics, Volcanism, and Climate Change. Pageoph Topical Volumes. Birkhäuser Basel

Love R., Milne G., Tarasov L, Engelhart S., Hijma M., Latychev K., Horton B., and  Tornqvis T. The contribution of glacial isostatic adjustment to projections of sea-level change along the Atlantic and Gulf coasts of North America, Earth's Future,4, 440-464, doi:10.1002/2016EF000363, 2016.

Webb, D. J.: On the tides and resonances of Hudson Bay and Hudson Strait, Ocean Sci., 10, 411-426, https://doi.org/10.5194/os-10-411-2014, 2014.

---

## Author Comment (AC2) · 19 Sep 2018

**General comments:**

While I can't comment on their statistical techniques, I noticed one major flaw in their approach. In calibrating radiocarbon ages of the shells the authors used the northern hemisphere terrestrial calibration curve - they are marine shells. They should be using the marine calibration curve and related to this would be the effects of marine reservoir corrections - surely this would impact the results and mean a reanalysis is necessary.

The first referee had the same comment, so we relate to the respective reply to Rev. 1. (cp-2018-50-AC1-supplement) on Page 5, comm. 8. Reanalysis with the new data has been done, and the results did not differ significantly.

From a writing perspective, the manuscript would benefit greatly from an improvement to the text. Notably, the descriptions of the indicative meaning of a sea level indicator used to create sea level index points/limiting constraints. I think there is some confusion here or at least it is not made clear. The introduction, in particular, could do with re-writing - there is a lot of unnecessary information and the goals of the paper should be made clearer, earlier on

We improved the text based on the suggestions of the referee and re-write certain parts of the introduction. We improved the description of indicative meaning and clear the confusion between sea-level indicators and sea-level index points. Therein, we followed the recommendations of Shennan et al. (2015, Chap 2). We included all the specific corrections into revised version of the manuscript.

**Specific comments:**

**Page 1**

L. 18    abbreviate here to RSL - you do this later on in the text but it is first used here.

Corrected.

L. 18-21 Poor English and vague statement. Have a better opening sentence.
Could be written so much better.
We rewrote the opening paragraph and tried to improve English.

L. 21    Elements??

Corrected term is features.

**Page 2**

L. 1    You're describing here features of the indicative meaning of a sea level index point. The SLI is used to establish a sea level index point. Refer to the sea-level handbook of Shennan 2015

We now described the difference between sea-level indicator and sea-level index point based on Shennan et al. (2015, Chap. 2).

"Fossil samples, morphological and archaeological features governed by the paleo sea level are defined as sea-level indicators (Shennan et al. 2015, van De Plassche 1986). Those sea-level indicators that are containing four main attributes: location of the sample on the Earth, deposition age, elevation related to present RSL, ordnance datum and the tendency, are expressed as sea-level index points, abbreviated to SLIPs or SLI (here, we will use SLI) and are commonly used to reconstruct past RSL ( Shennan et al 2015). Elevation of the SLI represents "indicative meaning" in relation to the present RSL (van De Plassche 1986). SLI living range with respect to the corresponding shoreline was introduced as indicative meaning by van de Plassche (1986) and contains two parameters: the reference water level and the indicative range. "

L. 2    Do they? Have any examples?

Here the aim was to point out that in order to differentiate SLIs from predictions and estimates of GIA models, they can be described as observations, while that is not correct term since SLIs have to be processed by e.g. dating methods, correction to GPS measurements or the statistical methods to obtain derivation of one of the 4 attributes (Shennan et al. 2015).

L. 3    establishing elevation of a SLI.
Note - leveling is a method from which we establish the elevation of SLI. You should use the correct terminology.

Corrected.

L. 3-4    models? do you mean techniques? i.e. c14 etc
if you are referring to age models they are usually restricted to continuous sequences of sediment, not SLI like shells.
the correct term would be modern analogue.

This sentence now reads as follows: " In addition to the elevation and age determination for each SLI, the relation to RSL has to be derived from the indicative meaning based on the modern analogue or the deposition conditions of the sample/specimen."

L. 3    This term should be introduced above with correct referencing of van de Plassche.

Done.

L. 4    i don't understand what your point is.
        This opening paragraph reads confused,

        As mentioned before, opening paragraph of the introduction, along with definitions has been
        rewritten to improve the understanding and goals of the paper.

L. 7    Examples?

        It is common to use 2σ range to define upper and lower limits. But, since we do not use this
        confidence interval, we removed this part of the sentence.

L. 7    Might want to introduction this definition earlier on. Also why start the sentence with "it is
        important to mention"?

        We introduced RSL further up.

L. 11-13 You introduced indicative meaning term above
        is all this info really needed in the introduction or paper at all? You could have one or two
        sentences max defining the indicative meaning, RSL etc with relevant references instead of
        the 2/3 paragraphs.

        This paragraph was reduced and we explained indicative meaning together with SLIs.

L. 25   I understand what you are saying but it is not written very well.
        Also note again - SLI are used to produce sea level index points. The SLIPs can be used to
        validate/tune GIA models.

        We are abbreviating sea-level index points as SLIs.

L. 25-35 This paragraph is too long and should be split in two. You talk about databases, GIA, stat
        methods. Break it up.

        Done.

L. 25   Such a vague statement. If you're going to say such things back it up with some references.

        This sentence was meant to introduce part that follows and describes different methods, and
        based on suggestion of the referee, that will be a new paragraph.

L. 26    You already abbreviated to RSL so why use in full. This is a simple thing to do.

Done.

L. 27    This should be a new paragraph.

Done.

L. 34    is respected the appropriate word??

We use this word as synonym to considered, concerned and we believe it is appropriate in this sense.

**Page 3**

Fig 1.    Low quality figure

We submitted better quality image.

[Figure]

**Figure 1.** Flowchart of model workflow.

L. 1    Hudson Bay, Canada....

Corrected.

L. 3    This is more suited to section below. Or remove completely.

This paragraph is placed in the section below.

L. 7    At last. This should be made clear in your introduction. What the aims of the paper are.

This part is shifted to the introduction.

L. 11     Is there a reference or web address for this data?

There is no reference for the data from GFZ database.

L. 15     Ref?

Data from Art Dyke is unpublished so there is no reference, we now placed "personal correspondence" next to it.

L. 16     I would prefer to see PDF not abbreviated in the headings.

We agree and corrected.

L. 17     Elevation

Corrected

**Page 4**

L. 4      What about other errors in height/elevation? like measurement or datum uncertainties?

The elevation of the SLIs is referenced to mean sea level. It was measured or it was taken from topographic maps; accordingly it is quite heterogeneous. Considered measurement errors in elevation from selected SLIs and OBIS data are explained in the Equations 5. and 6. We applied a uniform error of 1 m for OBIS data and 5 m for SLIs. Geographical positioning errors are not relevant for this study, therefore we do not consider them.

L. 13      Italics?

Set to italics letters.

L. 13     SO where do they typically live? what are their depth ranges? The reader is unlikely going to find this info from OBIS. Better to state.

Depth ranges of the selected shells are visible in the Figure 4. But we agree with the comment, and it is stated in the manuscript along with the information regarding their typical environment

**Page 5**

RSL at 800m??

The RSL as a correction term due to GIA is the difference between geoid displacement and radial surface displacement (Farell & Clark, 1976). Extended over land areas, it is dominated

here by the radial surface displacement due to the former ice load. Furthermore, we now show the distribution at 8 ka BP.

**Page 7**

Fig 5.   This is straight out of the OxCal software. At least make some effort to edit it yourself.

We generated new figure from the OxCal output. Considering the much smoother curve due to applying marine curve, we stay in using the PDF directly, as we aim to extend this method to terrestrial samples where the characteristics of the PDF will become more irregular.

[Figure]

**Figure 5.** Calibration curve generated from OxCal output with measured radiocarbon determination of 7010±75 BP (Ramsey, 2017). Marine curve "marine 13.14" was used in the calibration, with reservoir age correction (Delta_R) of 416±50 yr.

L. 2   You have to reference such statements - did you come up with this??

This paragraph has been rewritten with appropriate references.

"Radiocarbon technique has been used for dating of archaeological and geological records since its invention in the 1940s (Libby 1952, Reimer et al 2013). But $^{14}$C ages according to the decay of the radio nuclid do not represent directly calendar years due to the variable production rate of $^{14}$C in the upper atmosphere (Reimer et al 2103.). Determination of $^{14}$C age

is calculated by ratio of $^{14}C/^{12}C$, which depends on $^{14}C$ production and the conversion of $^{14}C$ age to calendar years is done with calibration curves (Törnquist et al 2015). Here we used marine curve "Marine13" (Reimer et al. 2013) and calibration software OxCal (Ramsay, 2017) to calibrate ages of considered SLIs. Since they are marine samples we had to take into account "reservoir effects". The $CO_2$ exchange between atmosphere and the ocean leads to a delayed uptake in the surface waters (e.g. Törnquist et al 2015). Due to this effect, called reservoir effect, marine samples will have lower concentration of $^{14}C$ than the terrestrial samples, having as a global average deviation of 400 $^{14}C$ years (Törnquist et al 2015). "

Furthermore, in reply to Rev 1., we considered a spatial variability of this correction (see P. 1, comm. 1).

L. 4    Not relevant

We removed this sentence.

L. 7    Why are you using this calibration?! They're marine shells. Also related are marine reservoir effects - did you not consider this? This would effect the ages and your results

See the answer to the comment Page 7, L. 2.

**Page 9**

L 2.    tense

Corrected

**Page 11**

L. 1    lithosphere

Corrected.

L . 4-6    Sentence doesn't read properly.

The sentence has been rewritten.

"In Table 2 , we present results of different studies that are estimating mantle viscosity for Hudson Bay region compiled in  Wolf et al. (2006) together with one further study and two global estimates. "

**Page 14**

L 2.-4    Awkward opening line.

Opening sentence is changed.

L. 8    You have used the wrong calibration curve. It should be marine.

See the answer to the comment Page 7, L. 2.

L. 10   no where in the paper did you say this until the conclusion…

This is corrected and added into the section 2.1 of the manuscript.

L. 15   First real mention of this. How would tidal range change effect your results? Moreover, how are the shells related to tides anyway??

Some shells are living in intertidal zone and can be effected by tides, but the tidal range for the Hudson Bay at present varies between 0 and 4 m (Webb 2013), and produces accordingly an offset depending on the living conditions with respect to the tidal range. We decided not to add this aspect as well as not to consider the relation of some samples to "storm-beaches" (see comment to Rev. 1. (cp-2018-50-AC1-supplement) P. 3, comm. 3)

**Literature**

Butzin, Martin; Köhler, Peter; Lohmann, Gerrit (2017): Marine radiocarbon reservoir ages for the past 50,000 years, links to model results in NetCDF format. PANGAEA, https://doi.org/10.1594/PANGAEA.876733, Supplement to: Butzin, M et al. (2017): Marine radiocarbon reservoir age simulations for the past 50,000 years. Geophysical Research Letters, 44(16), 8473-8480, https://doi.org/10.1002/2017GL074688

Farrell, W. E. and Clark, J. A. (1976), On Postglacial Sea Level. Geophysical Journal of the Royal Astronomical Society, 46: 647-667. doi:10.1111/j.1365-246X.1976.tb01252.x

Libby, W.F. (1952) Radiocarbon Dating, University of Chicago Press, Chicago

Reimer, P. J., Bard, E., Bayliss, A., Beck, J. W., Blackwell, P. G., Bronk Ramsey, C., Grootes, P. M., Guilderson, T. P., Haflidason, H., Hajdas, I., Hatt, C., Heaton, T. J., Hoffmann, D. L., Hogg, A. G., Hughen, K. A., Kaiser, K. F., Kromer, B., Manning, S. W., Niu, M., Reimer, R. W., Richards, D. A., Scott, E. M., Southon, J. R., Staff, R. A., Turney, C. S. M., , and van der Plicht, J.: IntCal13 and Marine13 Radiocarbon Age Calibration Curves 0–50,000 Years cal BP, Radiocarbon, 55, 1869–1887, https://doi.org/10.2458/azu_js_rc.55.16947, 2013

Shennan, I., Long, A. J., and Horton, B. P., eds.: Handbook of Sea-Level Research, Wiley, Blackwell, Chapter 2, 2015.

Törnqvist, T. E., Rosenheim, B. E., Hu, P. and Fernandez, A. B. (2015). Radiocarbon dating and calibration. In Handbook of Sea-Level Research (eds I. Shennan, A. J. Long and B. P. Horton). doi:10.1002/9781118452547.ch23

van de Plassche, O., ed.: Sea-level Research: A Manual for the Collection and Evaluation of Data, Norwich, Geo Books, Norwich, https://doi.org/10.1007/978-94-009-4215-8, 1986

Webb, D. J.: On the tides and resonances of Hudson Bay and Hudson Strait, Ocean Sci., 10, 411-426, https://doi.org/10.5194/os-10-411-2014, 2014.